# Intraductal Pancreatic Mucinous Neoplasms: A Tumor-Biology Based Approach for Risk Stratification

**DOI:** 10.3390/ijms21176386

**Published:** 2020-09-02

**Authors:** Vincenzo Nasca, Marta Chiaravalli, Geny Piro, Annachiara Esposito, Lisa Salvatore, Giampaolo Tortora, Vincenzo Corbo, Carmine Carbone

**Affiliations:** 1Department of Medical Oncology, Fondazione Policlinico Universitario Agostino Gemelli IRCCS, 00168 Rome, Italy; nascavincenzo@gmail.com (V.N.); marta.chiaravalli@gmail.com (M.C.); genypiro@hotmail.com (G.P.); annachiara.esposito@policlinicogemelli.it (A.E.); lisa.salvatore@policlinicogemelli.it (L.S.); giampaolo.tortora@policlinicogemelli.it (G.T.); 2Medical Oncology, Università Cattolica del Sacro Cuore, 00168 Rome, Italy; 3Department of Diagnostics and Public Health, University of Verona, 37134 Verona, Italy; vincenzo.corbo@univr.it; 4ARC-Net Research Centre, University of Verona, 37134 Verona, Italy

**Keywords:** IPMN, pancreatic cancer, carcinogenesis, pancreas

## Abstract

Pancreatic ductal adenocarcinoma is one of the most lethal human cancers. Its precursor lesions include pancreatic intra-epithelial neoplasia, mucinous cystic neoplasm, and intraductal papillary mucinous neoplasm (IPMN). IPMNs usually present as an incidental finding at imaging in 2.6% of the population and, according to the degree of dysplasia, they are classified as low- or high-grade lesions. Since the risk of malignant transformation is not accurately predictable, the management of these lesions is based on morphological and clinical parameters, such as presence of mural nodule, main pancreatic duct dilation, presence of symptoms, or high-grade dysplasia. Although the main genetic alterations associated to IPMNs have been elucidated, they are still not helpful for disease risk stratification. The growing body of genomic and epigenomic studies along with the more recent development of organotypic cultures provide the opportunity to improve our understanding of the malignant transformation process, which will likely deliver biomarkers to help discriminate between low- and high-risk lesions. Recent insights on the topic are herein summarized.

## 1. Introduction

Pancreatic ductal adenocarcinoma (PDAC) is one of the most lethal human cancers, with a 5 year-overall survival of 9% [1]. The dismal prognosis is mainly contributed by the lack of specific symptoms of early stage disease, the absence of screening programs and the paucity of effective therapeutic strategies [2]. Accordingly, efforts are being made towards elucidating mechanisms of carcinogenesis through the study of pancreatic cancer precursor lesions, which include pancreatic intra-epithelial neoplasia (PanIN), being the most common, mucinous cystic neoplasm (MCN), and intra-ductal papillary mucinous neoplasm (IPMN). MCNs are rare, slow-growing cystic tumors, primarily affecting women, that do not arise from the ductal system of the pancreas and are characterized by an uncertain potential for progression to PDA [3,4]. PanINs can be classified into a three-grade system according to the degree of dysplasia, where PanIN3 (presenting high-grade dysplasia) is often associated with carcinoma [5,6]. The PanIN to pancreatic cancer progression model has been well investigated and the associated genetic events identified; however, being microscopic lesions (usually < 5 mm), they cannot be detected by imaging modalities (differently from MCNs and IPMNs) [7]. In contrast, the biology of IPMNs is less well understood, albeit these macroscopic entities (usually > 5 mm) can be detected in up to 2.6% of the population as incidental findings [8]. Furthermore, PanINs and IPMNs have overlapping yet distinct genetic aberrations (e.g., *KRAS* mutations), while other alterations are unique to IPMNs (e.g., *GNAS*; genetics of IPMN is discussed below) [9,10].

Indeed, the rate of malignant transformation of IPMN into PDAC is highly variable and the mechanisms driving progression are still unclear [10]. Current guidelines for the management of PDAC precursor lesions are based merely on clinical and radiological parameters [11,12,13], which cannot discriminate the degree of dysplasia of the lesions. Limitations to this approach are, on one hand, the failure to accurately detect high-risk disease and on the other the high rate of unnecessary surgery for low-risk IPMNs [14]. In this scenario, the development of molecular biomarkers predictive of malignant behavior is warranted. Herein, we focus on the state-of-the-art and future insights into IPMN management and research.

## 2. Definition, Epidemiology, and Classification Systems

The first recognition of IPMN as a different entity of pancreatic cystic disease dates back to 1986 when Itai et al. described a small series of pancreatic cystic neoplasms (PCNs) with localized cystic dilations, classifying them as “ductectatic mucinous cystadenoma and cystadenocarcinoma” [15]. What distinguished them from the already-known PCNs was the involvement of the major pancreatic duct and the absence of an ovarian stroma. This very first description does not fall far away from the current definition of IPMN: a grossly visible (>5 mm) intraductal epithelial neoplasm of mucin producing cells arising in the pancreatic ducts [16]. However, it was not until 1996 that IPMNs were officially recognized by the WHO as a separate subset of pancreatic neoplasms [17]. Due to the widespread use of high-quality imaging the incidence of PCNs is increasing with estimates of incidental cysts varying between 9% and 41% and depending on the type of imaging modality used [18,19,20]. However, the true prevalence of IPMNs remains elusive. They usually present in the fifth to seventh decades of life with an equal gender distribution. The malignant potential is higher, ranging between 36% and 100%, when the main pancreatic duct is involved; on the contrary, when the branch ducts are involved the malignant potential is about 15%. Of interest, a causative link between IPMN and diabetes mellitus has been described, and 10–45% of patients with a diagnosis of IPMN have concomitant diabetes mellitus [21].

Current classifications of IPMNs are based on morphological features detected at imaging and used in the clinical practice and histological properties (Table 1).

Morphologically, they are classified as main duct (MD-IPMN), branch duct (BD-IPMN), or mixed type (MT-IPMN). This classification is related to the risk of malignancy with main duct involvement being a sign of high-risk disease. Indeed, in resected specimens of BD-IPMN, high-grade dysplasia (HGD) or invasive carcinoma were found in 31% and 18.5% of the cases, while in resected MD-IPMNs the percentages of HGD and invasive carcinoma rose up to 61.6% and 41.3%, respectively [13].

IPMNs can typically progress into two distinct malignant tumors, namely tubular and colloid. Tubular carcinomas resemble classic PDAC and possess a similarly dismal prognosis, while colloid carcinomas present with abundant extracellular mucins and interspersed cancerous epithelium and, overall, possess a more favorable prognosis [22,23,24,25,26,27].

Histologically, IPMNs can be grouped according to the pattern of mucins expression in the following subtypes: gastric, which is similar to gastric foveolar cells and expresses MUC5AC; intestinal, which shows similarity with colonic epithelium and expresses CDX2 and MUC2; and pancreaticobiliary, which expresses MUC1 and MUC5. Interestingly, BD-IPMNs usually show gastric histology, which has a lower likelihood of progression to invasive cancer. However, when they do progress, BD-PIMNs present with a tubular/ductal carcinoma histology, which, as mentioned above, has a very poor prognosis, similar to classic PDAC [28]. MD-IPMNs more often show intestinal and pancreaticobiliary histology, which tend to be high-grade lesions and are at increased risk of progression to invasive carcinoma [26,29,30]. Of note, intestinal IPMNs give rise to colloid carcinomas, while tubular carcinomas usually arise from gastric and pancreaticobiliary IPMNs [25,31].

Oncocytic IPMNs, previously classified as a fourth histological subtype, are now recognized as a separate entity by the 2019 WHO classification [16], as histology shows complex papillae with cuboidal lining cells and have only focal expression of mucins; they are usually BD-IPMN and often present with HGD or invasive carcinoma [32]. When they progress into invasive cancer, they give rise to oncocytic or tubular carcinoma.

Furthermore, based on the degree of dysplasia, the 2015 Baltimore Consensus Meeting issued recommendations for pancreatic precursor lesions, leading to a two-tier system grouping low- and high-grade lesions, leaving behind the previous intermediate-grade dysplasia [5].

## 3. State-of-the-Art Management

Diagnosis of IPMN is usually achieved via computed tomography (CT), magnetic resonance imaging (MRI), and endoscopic ultrasound (EUS) [33,34,35], which showed to be accurate in detection of cystic lesions but do not allow to discriminate the grade of dysplasia of the neoplasm, and, therefore, the ultimate likelihood of progression to invasive carcinoma. This distinction is of paramount importance because of the heterogeneity of the spectrum of pancreatic cystic disease, which also comprehends lesions that will never have a malignant transition [36].

In 2006, the American Pancreatic Association published consensus guidelines regarding IPMN management called the *Sendai Criteria* [37], which were adjusted into the *Fukuoka Guidelines* in 2012 and subsequently updated in 2017 [13]. These guidelines restricted indications for surgery in the presence of “high-risk stigmata” of malignancy, such as large mural nodules, marked dilation of the main pancreatic duct (>10 mm), positive cytology for HGD, or obstructive jaundice. Further investigations and follow-up are reserved for lesions with “worrisome features” such as cyst >30 mm, main pancreatic duct dilation 5−9 mm in size, pancreatitis, non-enhancing nodules, atypical cells at cytology, or minor symptoms. In 2013, the European Study Group on Cystic Tumors of the Pancreas, also provided consensus guidelines in this setting, however they were based on expert’s review of the literature [38]. This prompted the American Gastroenterology Association (AGA) [11] in 2015 and the European Study Group [12] in 2018 to perform a systematic review of the literature and to update the guidelines. Notably, as IPMNs are frequently detected incidentally, surgery can be considered only in fit patients taking into account an individual’s life expectancy. Indications for surgery are summarized in Table 2.

Despite the numerous recommendations, the features that are evaluated might still be insufficient to correctly identify the best management strategy for patients. For example, the presence of mural nodules is highly related to the histological grade of the lesion [39,40], however according to Karasaki and colleagues, mural nodules were found within the most dysplastic lesion only in 21.1% of IPMNs [41]. Malignant progression is therefore not limited to mural nodules, and flat lesions may need to be surveilled as well for their potential to progress to invasive carcinoma [42]. Recently, a study conducted on 295 patients concluded that the presence of enhancing mural nodule ≥ 5 mm (odds ratio, OR: 4.1), pancreatitis (OR: 2.2), and thickened/enhancing cyst walls (OR: 2.2) were independent predictive factors of HGD [43]. Attention has been now directed toward EUS-guided techniques. Cyst fluid analysis for biochemical markers and atypical cells (even if the fluid is mostly acellular and therefore of limited use) and confocal laser endomicroscopy currently lead the field in risk stratification of IPMNs. Encouraging results need validation in larger studies, and not always current procedures led to accurate detection of invasive carcinoma, but these methods seem a promising road for the early detection of cancer. Confocal laser endomicroscopy showed to be superior to cyst fluid CEA and cytology, and new parameters (e.g., papillary epithelial width, darkness) may be evaluated to detect high-grade dyplasia in IPMNs [44,45,46,47,48]. For example, evaluation of cyst fluid CEA may differentiate between mucinous and non-mucinous cystic neoplasm (CEA > 192 ng/mL suggests an MCN), yet it is not helpful in distinguishing MCNs versus IPMNs or benign versus HGD lesions [45,49,50,51,52,53,54,55,56]. Following cell death, genetic materials are released into the cystic fluid making the detection of IPMN-associated DNA mutations possible. Indeed, testing for common IPMN mutations (e.g., *KRAS* and *GNAS*) in cystic fluid achieved 96% sensitivity and 100% specificity in making a diagnosis of IPMN. However, detection of those mutations cannot differentiate low-grade lesions versus HGD [57].

The so-called “field defect” further complicates the IPMN management and virtually poses the entire pancreas at risk for malignancy [58]. This was also suggested by noticing during follow-up the occurrence of PDAC distant and separate from the original IPMN, at a rate of 7−8% [59]. Although this rate is lower than the one describing invasive carcinoma arising from IPMN, it must be taken into account in the comprehensive assessment and management of these patients. Another feature is the relatively frequent multifocal presentation of the disease (5−8% of the cases), either synchronously or metachronously [29,60,61]. For these reasons, a follow-up program after surgery is mandatory [13,62].

Of note, the Italian Association of Hospital Gastroenterologists and Endoscopists (AIGO) and the Italian Association for the Study of the Pancreas (AISP), are currently involved in a survey to prospectively validate their guidelines, which foresees surgery in the presence of an enhancing solid component within the cyst or main duct > 10 mm in asymptomatic patients [63]; preliminary data are encouraging in producing a reliable real-life diagnostic workup and management of PCNs [64].

## 4. Genetics and Molecular Pathways

Several studies identified the main molecular features and the heterogenous pathway of progression that characterize the biology of IPMNs [10,65].

The most commonly found genetic alterations in IPMNs are *KRAS* and *GNAS* mutations [66,67], which were found to cooperate in promoting pancreatic tumorigenesis in animal models [68].

The role of *KRAS* in PDAC and other human cancers is well established [69]; mutations in this gene are an early event during IPMN development and are present in up to 80% of the cases [5,70,71,72]. A recent study confirmed the synergistic action of *KRAS* and tumor suppressor gene mutations for development of IPMN in animal model and highlighted the role of Wnt/β-catenin pathway in *KRAS*-associated lesions [73].

*GNAS* mutations are found in around 70% of IPMNs and are absent in other precursor lesions or in invasive PDAC not associated to IPMN [66,74,75]. *GNAS* mutations lead to a constitutionally activated G-protein α-subunit which in turns activates the cyclic-AMP cascade to promote cell growth and proliferation [76]. Although they can be found in any IPMN subtype, *GNAS* mutations are more frequently observed in the intestinal subtype [77,78]. Interestingly, *GNAS* has also been shown to harbor tumor-suppressor gene abilities in other tumors [79,80], suggesting that the output of *GNAS* mutations is highly contextual in cancers [63,64].

Recently, three studies attempted to unveil the molecular mechanisms of progression of early lesions. Omori et al. firstly identified three pathways of progression from IPMNs to PDAC by assessing the clonal relatedness of concurrent lesions: “sequential”, “branch-off”, and “de novo” [81]. The “sequential” subtype progresses to invasive cancer via an accumulation of mutations in a stepwise manner, inheriting completely the *KRAS* and *GNAS* signature of the starting IPMN. The other two subtypes instead undergo different pathways: PDAC can arise from a common founder clone coexisting in the IPMN (“branch-off”) or from a completely independent clone (“de novo”).

Fischer et al. investigated the genetic evolution IPMN by multiregional sequencing of early- and late-stage lesion, showing substantial heterogeneity in driver mutations across early lesions, which contained multiple independent clones, and the lack of this heterogeneity in high-grade IPMNs. This suggests that only one of the multiple clones undergoes selection and expansion, after acquiring mutations in later driver genes [82].

*RNF43* loss-of-function mutations are found in about 75% of IPMNs [66,67]. The *RNF43* product inhibits cell proliferation by exerting a negative regulatory action on the Wnt signaling [83]. As *RNF43* mutations have been identified in a variable proportion of PDACs [84,85], it will be worthwhile to further analyze its role in cancer maintenance for therapeutic purposes. Of interest, in vivo studies highlighted an increased sensitivity of PDAC harboring *RNF43* mutations to Wnt-pathway pharmacological inhibitors, which may represent a new option for Wnt-addicted subset of PDAC [86,87]. Noë et al. recently showed that distinct *RFN43* alterations are present in multiple precancerous clones, but are lacking in invasive cancer, suggesting that some mutations characterizing early noninvasive lesions are not selected for invasive carcinoma [72].

Other relevant mutations found in IPMNs involve *CDKN2A*, *SMAD4*, and *TP53* genes, which are the main mutations also present in PDAC. Target sequencing studies clarified that mutations affecting these genes are predominantly found in high-grade lesions and invasive carcinomas with respect to low-grade IPMNs (Table 3).

*CDKN2A* alterations include inactivating mutations, epigenetic silencing and chromosome 9p loss [19,20,41]. Next-generation sequencing (NGS) studies demonstrated a prevalence of 0−2% of *CDKN2A* mutations in low-grade IPMNs and up to 15% in high-grade IPMNs [67].

*SMAD4* mutations are rarely observed in IPMNs while enriched in invasive carcinomas, thereby suggesting that they are acquired during late stages of tumorigenesis; the hypothesis that alterations involving *SMAD4* may drive malignant transformation and invasion in a subset of lesions is supported by a recent work by Noë et al. [66,67,72,88]. In order to be translocated into the nucleus, Smad4 has to be bound to phosphorylated Smad3, which can be phosphorylated as follows: COOH-terminally phosphorylated Smad3 (pSmad3C), frequently detected in normal pancreatic tissues, linker-phosphorylated Smad3 (pSmad3L), rarely found in normal tissues, and Smad3 phosphorylated at both sites (pSmad3L/C). The expression of pSmad3C was confirmed in low-grade IPMNs and was less frequent in HGD IPMNs; the contrary was true for the immunopositivity of pSmad3L. Indeed, immunostaining index for pSmad3C was 79.2% in low-grade, 74.9% in HGD and 42% invasive cancer; whereas the index of pSmad3L was 3.4% in low-grade, 4.3% in HGD and 42.4% in invasive cancer respectively, and these differences were statistically significant. Therefore, signal alteration in pSmad3 seems involved in the carcinogenesis of IPMN and the pSmad3L/pSmad3C ratio could represent a novel biomarker to predict recurrences in patients who undergone surgery [89].

Similar to *SMAD4*, *TP53* mutations are frequently present in high-grade lesions and very rarely seen in low-grade IPMN [67,78]. This prompted to retrospectively investigate the role of serum anti-p53 antibody (S-p53Ab), an auto-antibody produced against aberrant *p53* expression, as a predictor of HGD in IPMNs. S-p53Ab was absent in low-grade dysplasia IPMNs, and its incidence increased with the level of dysplasia (5.3% in HGD-IPMNs and 11.4% in IPMNs with foci of invasive cancer). However, in this study, only six patients expressed S-p53Ab with very low sensitivity (8.2%), thus limiting the conclusions that may be drawn [90].

Other recently discovered novel driver genes may contribute to IPMN tumorigenesis. Both germline and somatic *ATM* mutations have been found in up to 17% cases; other newly proposed driver genes are *GLI3* and *SF3B1* [72].

In summary, further studies could provide insights into transition of IPMN to invasive carcinoma [72,81,82], with mutations in *CDKN2A*, *SMAD4*, and *TP53* genes probably not fundamental in the early evolution of IPMNs. Recently, Bayesian hierarchical models to study the evolutionary timeline of high-grade IPMN to PDAC estimated that at least three years are needed for acquisition of invasive properties: this significant time window provides us with an opportunity for surveillance and early detection of invasive cancer; it will therefore be of uttermost importance to find the correct strategy to pursue this goal [72].

A different matter concerns the genetics of oncocytic IPMNs, which were less studied with respect to the other subtypes and, as previously mentioned, are a distinct entity with unique features. Notably, a targeted sequencing study of nine oncocytic IPMNs showed no mutations in *KRAS* and *GNAS* genes, and *RNF43* mutation appeared only once [32]. Another more recent study, found the presence of recurrent fusions of *PRKACA* and *PRKACB* genes in all the 23 surgically resected oncocytic IPMNs under examination. The same samples were negative for the key genomic alterations found in other pancreatic neoplasms, supporting the distinct entity of oncocytic IPMNs [91]. It will be important to confirm these results in larger studies and possibly explore different cancer gene panels for oncocytic IPMNs.

## 5. IPMN Microenvironment

PDAC tissues have a unique and abundant peri-tumoral stroma composed by extra-cellular matrix proteins, fibroblasts, endothelial cells, pericytes, and immune cells that collectively shape the biological behavior of the tumor as well as response to therapeutic approaches. Overall, the pancreatic tumor microenvironment has an immunosuppressive phenotype, which is dominated by myeloid cells and with rare cytotoxic T lymphocytes. In contrast, IPMNs are found to harbor antitumor immune components [92]. Indeed, major changes in the immunological landscape from low-grade IPMNs to invasive carcinoma pertain the expansion of immune suppressive FOXP3+ T-regs and CD68+ macrophages at the expense of CD8+ T lymphocytes. This suggests that the change in T-cells composition and the loss of immune surveillance mainly occurs in the progression from high-grade IPMN to PDAC. Other immune cells that can be found in the tumor microenvironment are tumor-associated neutrophils (TANs), that have been previously associated to HGD-IPMNs [93]. A retrospective study confirmed that 96% of low-risk lesions were TAN negative and 89% high-risk lesions had high levels of TANs. The authors proposed the presence of inflammatory markers in the cyst fluid as a surrogate marker for TAN. Indeed, they found that high TAN is correlated to high levels of cyst fluid inflammatory markers (i.e., TNFα, IL-4, IL-1β, IL-α, INF-γ, IL-2) [94].

## 6. Recent Attempts to Predict Malignancy

The first single-cell transcriptomic profiling of PDAC precursor lesions performed to understand the progression from low-risk IPMN to invasive cancer was published in 2018 by Bernard and colleagues [95]. They identified 10 clusters of unique stromal and epithelial components. Signatures of pancreatic epithelium, such as *MUC1* and *KRT19,* were present in all the lesions, while *CEA-CAM6* and *MUC5AC* expression was enriched in HGD/PDAC samples and low-grade IPMNs, respectively. Low-grade lesions expressed tumor-suppressor genes, which were deregulated in HGD where a higher expression of oncogenic transcripts was seen. A significant finding was the high proportion of myeloid-derived suppressor cells within the stroma of PDAC compared to low-grade and HGD-IPMNs. Myofibroblasts (MyCAFs) were rare in low-grade IPMN, but highly represented in HGD-IPMNs. In contrast, inflammatory fibroblasts (iCAFs) were only identified in the PDAC samples. Collectively, these data suggest that quality other than quantity of stromal cells might be informative of an aggressive phenotype.

The role of pre-operative serum CA 19.9 and CEA, which are known to be markers of PDAC, was evaluated in 594 resected patients [96]. An elevated CA 19.9 was more likely found in patients with invasive carcinoma at histology, in particular 63% of patients with above normal CA 19.9 levels had either HGD or invasive cancer, but the increase in serum marker is not independently associated to HGD as about 72% of these patients had a normal tumor marker. Conversely, elevations of serum CEA were not associated with risk of malignancy in IPMN.

Serum proteome was retrospectively investigated by a microarray platform in a cohort composed by 56 IPMN patients [97]. The combination of six serum proteins (MUC17, ID3, AREG, ITGA2B, CSF2RA, and CCR5) together with clinical parameters (levels of bilirubin and size of the cyst) provided a predictive signature of HGD with a 93% accuracy rate. However, given the retrospective nature and the small sample size of this analysis, further studies should validate these findings, which are of interest as implied a non-invasive and easily accessible tool.

Global and gene-level changes in DNA methylation have been observed during progression of several epithelial tumors, including colorectal cancer [98]. In a recent work, Fujiyama and colleagues [99] investigated whether DNA methylation level of the *Cysteine Dioxygenase 1* (*CDO1*) gene could be useful in discriminating between benign (low- and intermediate-grade dysplasia) and malignant (HGD and invasive carcinoma) IPMNs. *CDO1* hypermethylation was found to be a predictor of malignancy with a sensitivity of 85.4% and a specificity of 66.7%, and its hypermethylation increases during the adenoma–carcinoma progression in IPMNs. Again, results must be validated in a larger cohort of patients and a cut-off value is still to be proposed.

A less explored field of PDAC biology is the role of long non-coding RNAs (lncRNAs) in tumor progression. Permuth et al. demonstrated that a signature composed of eight lncRNAs could help in permitting an accurate differential diagnosis between malignant and non-malignant IPMNs [100]. Moreover, three lncRNAs (HAND2-AS1, CTD-2033D15.2, and lncRNA-TGF) were found to be associated with IPMN tumorigenesis and, if their role will be validated, they could potentially be exploited as early diagnostic biomarkers [101].

Due to their stability in body fluids, detection of MicroRNAs (miRNAs) in liquid biopsy has been evaluated as a tool to help differentiate between pancreato-biliary cancer histotypes. Exosomal MiRNA profiling with a digital detection technology found 54 deregulated miRNAs between IPMN and PDAC and 21 upregulated miRNAs in IPMN compared to PDAC. In particular, miR-17-5p and miR-106-5-p were significantly upregulated in carcinomas of the Ampulla and in IPMNs. Moreover, 12 miRNAs were found to be deregulated between malignant and non-malignant IPMNs [102].

Overall, further studies are needed to elucidate the genetic heterogeneity and polyclonal evolution of precancerous lesions.

## 7. Future Perspectives: Development of Organoid Model

Preclinical models of human IPMNs are rare and those will be a valuable tool to validate the biological significance of genetic and non-genetic alterations identified through the genome-wide analysis conducted to date. The application of the organoid technology to the pancreas has finally made possible to grow non-neoplastic and preneoplastic cells from pancreatic tissues [103,104]. More recently, organoid models were generated from IPMNs (10 samples) and from normal pancreatic ducts (7 samples) of patients who had undergone pancreatic resection at the Johns Hopkins Hospital [105]. Whole genome sequencing of these models identified 16 genes with recurrent non-silent somatic mutations in ≥2 IPMNs, namely *KRAS*, *GNAS*, *RNF43*, *CYP4Z1*, *DNAH9*, *HLA-DQB2*, *KIAA1109*, *MUC4*, *MUCC12*, *PHF3*, *RBM10*, *RXFP2*, *SLC7A8*, *SLC9A3, ZNF260*, and *ZNF835*. Overall, the most frequently mutated genes in IPMNs were *KRAS*, *GNAS*, and *RNF43*. This was in line with the current literature. Twenty-eight genes were aberrantly expressed at RNA-sequencing analysis in IPMNs with respect to normal duct organoids. The most significantly upregulated gene was *CLDN18*, while the most significantly downregulated one was *FOXA1*. Accordingly, *FOXA1* expression was decreased at immunohistochemistry. In particular, its expression was significantly decreased in low-grade IPMNs compared to normal duct (*p* < 0.0016).

Recently, our group has as well established and characterized organoid models from two patients affected with IPMN with an associated PDAC and local lymph node dissemination. We are currently expanding the array of IPMN-derived models in order to compare malignant and non-malignant IPMNs by whole genome sequencing and RNA-sequencing. The final aim is to identify and validate biomarkers to discriminate between malignant and non-malignant IPMNs.

## 8. Conclusions

We here reviewed management guidelines for IPMNs and the current knowledge on IPMN’s biology and behavior. As discussed above, the mere radiological and clinical signs are not sufficient in guiding management and follow-up of patients with IPMNs, and the identification and validation of molecular biomarkers predictive of malignancy are an important unmet need. NGS-techniques and the rising expertise in the field of organotypic cultures [103] are promising tools in this setting. Biomarkers to predict malignancy are urgently needed and will allow clinicians to adopt the correct management option, balancing the benefits of therapeutic intervention and/or active surveillance in high-risk patients and will reduce the risk of over-treatment. Moreover, it would allow diagnosis of PDAC at very early stages, improving the patient’s overall survival.

## Figures and Tables

**Table 1 ijms-21-06386-t001:** Differences between subtypes of IPMNs.

	GASTRIC	INTESTINAL	PANCREATICOBILIARY
Prevalence	Most Common (70%)	Second most Common (20%)	Least Common
Mucins	*MUC5AC*	*MUC2*	*MUC1*
	*MUC6*	*MUC4*	*MUC5AC*
		*MUC5AC*	*MUC6*
		*CDX2*	
Histology	Similar to gastric foveolar cells;	Similar to colonic epithelium;	Complex thin branching papillae
	finger like papillae	villous papillae	
Associated to	BD-IPMN	MD-IPMN	MD-IPMN
Risk of Malignancy	Low (10%)	High (40%)	High (68%)
Evolution in Cancer	Tubular carcinoma	Colloidal carcinoma	Tubular adenocarcinoma

**Table 2 ijms-21-06386-t002:** Comparison between the most recent guidelines for the indications to surgery.

	AGA Guideline 2015 [11]	Fukoka Guideline 2017 [13]	Revised EU Guideline 2018 [12]
Parameters	NA	High risk stigmata: -Enhancing mural nodule ≥ 5 mm; -MDP > 10 mm; jaundice. Worrisome features: -Growth ≥ 5 mm/2 years; -Cyst size ≥ 3 cm; -Enhancing mural nodule <5 mm; -Enhanced thickened cyst wall; -MDP 5−9 mm; -PD calibre change; -Elevated serum CA 19.9; -Pancreatitis.	Absolute indications: -Solid mass; -Enhancing mural nodule >5 mm; -MPD > 5 mm; -HGD/carcinoma; -Jaundice; -Positive cytology for malignancy/HGD. Relative indications: -Cyst growth rate ≥ 5 mm/year; -MPD dilation between 5 and 9.9 mm; -Cyst size ≥ 4 cm; -Enhancing mural nodule < 5 mm; -Serum CA 19.9 ≥ 37 U/mL -New onset DM; -Acute pancreatitis.
Indications for Surgery	Solid component and dilated MPD and/or concerning features on EUS-FNA	≥ 1 high risk stigmata; ≥ 1 worrisome feature and ≥ of the following: -Definite mural nodule; -MPD involvement; -Suspect cytology. Consider in: Cyst >2 cm in young and fit pts	≥ 1 absolute indication; ≥ 1 relative indication without comorbidities; ≥ 2 relative indications in pts with significant comorbidities.

MPD, main pancreatic duct; PD, pancreatic duct; HGD, high grade dysplasia; DM, diabetes mellitus; NA, not applicable.

**Table 3 ijms-21-06386-t003:** Rate of mutations in low and high-grade IPMN

	Low-Grade IPMN	High-Grade IPMN
***KRAS***	43−89%	31−71%
***GNAS***	41−77%	42−72%
***RNF43***	10%	25−75%
***CDKN2A***	<5%	0−15%
***TP53***	<5%	18−20%
***SMAD4***	<5%	<5%

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
