# Peer review of "Intraductal Pancreatic Mucinous Neoplasms: A Tumor-Biology Based Approach for Risk Stratification"

_ijms, 2020, doi:10.3390/ijms21176386_

Round 1

Reviewer 1 Report

This is a concise, timely, and well written review that summarizes an important topic and should be accepted for publication.

The only minor point for revision would be some moderate English copyediting and formatting:

  • discriminate (line 26)
  • cancers (line 31)
  • disease (line 33)
  • state-of-the-art (line 86)
  • materials are or material is (line 122)
  • add "a" before diagnosis (line 124)
  • take out "the" before IPMN (line 127)
  • have (line 217)
  • add "the" before pancreatic (line 219)
  • organoid (lines 280 and 285)
  • are (line 305)
  • The sentence beginning on line 307 is missing a word between "allow" and "to"
  • "stages, improving patients' " instead of "stages improving the patient's" (line 310)

I also thought Table 1 could be reformatted so it is easier to read- perhaps adding lines or bullet points to separate different features and guidelines

Author Response

Reviewer #1:

This is a concise, timely, and well written review that summarizes an important topic and should be accepted for publication.

The only minor point for revision would be some moderate English copyediting and formatting:

  • discriminate (line 26)
  • cancers (line 31)
  • disease (line 33)
  • state-of-the-art (line 86)
  • materials are or material is (line 122)
  • add "a" before diagnosis (line 124)
  • take out "the" before IPMN (line 127)
  • have (line 217)
  • add "the" before pancreatic (line 219)
  • organoid (lines 280 and 285)
  • are (line 305)
  • The sentence beginning on line 307 is missing a word between "allow" and "to"
  • "stages, improving patients' " instead of "stages improving the patient's" (line 310)

R: Thank you for your comments and corrections, we made all the copyediting and formatting changes requested.

I also thought Table 1 could be reformatted so it is easier to read- perhaps adding lines or bullet points to separate different features and guidelines

R: Copyediting and formatting modifications of Table 1 (Table 2 in the new version) have been reported as suggested.

Reviewer 2 Report

I carefully read the review article ": Intraductal Pancreatic Mucinous Neoplasms: a tumor-biology based approach for risk stratification" by Naca and coworkers, givingn an overview on  IPMN, cystic pancreas tumors, which can progress into invasive cancer. Although many issues are discussed I have some points to be clarified.

1.) Does the manuscript fulfill the scope of IJMS, where more molecular science, rather than clinical/pathological reviews are published. If yes, maybe the molecular alterations should be emphasized much more detailed

2.) Introduction: the Definition of IPMN is clear according to the new WHO. mucinous intraductal tumors > 5 mm. Please use the correct definition and not old concepts. (see WHO classification of tumors of the digestive tract).

3.) Elaborate the differences of the 3 histological subtypes. According to the new WHO the old oncocytic subtype is an own tumor entity. Please comment on this. 

4.) A table with the histological and molecular differences for the subtypes should be included, also with clear histological pcitures.

5.) Please discuss the differences and similarities of PanIN vs IPMN. (molecular phenotypes, predicitive values)

6.) Please include a chapter on Features of  different types of the IPMN associated cancers (PDAC vs. colloid carcinoma)

7.) Please comment on the diagnostics of IPMN: radiology, cyst fluid, molecular alterations in cystic fluid, tumor DNA, as many new advances were done.

8.) be aware: main duct and branch duct type are clinical and not pathological subtypes.

Author Response

Reviewer #2:

I carefully read the review article ": Intraductal Pancreatic Mucinous Neoplasms: a tumor-biology based approach for risk stratification" by Naca and coworkers, givingn an overview on  IPMN, cystic pancreas tumors, which can progress into invasive cancer. Although many issues are discussed I have some points to be clarified.

1.) Does the manuscript fulfill the scope of IJMS, where more molecular science, rather than clinical/pathological reviews are published. If yes, maybe the molecular alterations should be emphasized much more detailed

R: Thank you for bringing up this point. As suggested, we updated the “Genetics and molecular pathways” section with the very recent work by Noë et al. “Genomic characterization of malignant progression in neoplastic pancreatic cysts” (Nat Commun, 14 August 2020) that currently represents the largest dataset of whole exome sequencing of IPMNs; not only the results confirmed the idea that IPMNs (and MCNs) are direct precursors of invasive pancreatic cancer, but they also revealed novel driver mutations (e.g. ATM), clarified how some mutations are important for early tumorigenesis but not furtherly selected for the invasive process (e.g. RNF43), while others are likely contributors to this important malignant process (e.g. SMAD4).

 However, what we want to highlight with our paper is how little we know about molecular biology of IPMNs: currently, IPMN is a clinical problem and is almost exclusively managed with clinical criteria only. Major breakthroughs into their tumorigenesis and malignant transformation are still lacking, and we end our discussion proposing organoid model as a valid alternative (already deployed to study other cancerous lesions) to study its biology and, eventually, come up with solutions to guide clinicians in their clinical practice.

2.) Introduction: the Definition of IPMN is clear according to the new WHO. mucinous intraductal tumors > 5 mm. Please use the correct definition and not old concepts. (see WHO classification of tumors of the digestive tract).

R: We apologize for the mistake, we have corrected the text accordingly at lines 67-68.

3.) Elaborate the differences of the 3 histological subtypes. According to the new WHO the old oncocytic subtype is an own tumor entity. Please comment on this. 

R: We thank the reviewer for the opportunity to better clarify the differences of the subtypes of IPMNs, highlighting the new entity of intraductal oncocytic papillary neoplasm. We also provided a table that summarize the features of the histological subtypes (as suggested as well at point 4).

4.) A table with the histological and molecular differences for the subtypes should be included, also with clear histological pcitures.

R: We thank the reviewer for the suggestion. Table 1 has been included in text.

5.) Please discuss the differences and similarities of PanIN vs IPMN. (molecular phenotypes, predicitive values)

R: Thank you for the suggestion, we included in the introduction brief definitions of PanIN and mucinous cystic neoplasms, and a short comparison of PanIN vs IPMN.

6.) Please include a chapter on Features of  different types of the IPMN associated cancers (PDAC vs. colloid carcinoma)

R: We now included comments on the different types of cancers associated with the subtypes of IPMNs both in text (lines 97, 106, 110) and in Table 1.

7.) Please comment on the diagnostics of IPMN: radiology, cyst fluid, molecular alterations in cystic fluid, tumor DNA, as many new advances were done.

R: Thank you for this advice, which was in line with other reviewer’s comment. We accordingly updated the diagnostic and management session with recent steps forward, yet underlining that larger studies are needed to validate the early promising results cited. 

8.) be aware: main duct and branch duct type are clinical and not pathological subtypes.

R: We better clarified this concept in line 79. With the term “morphological” we do not mean “pathological”. Indeed the distinction in “main” or “brunch duct” is made at imaging and is used in clinical practice.  

Reviewer 3 Report

REVIEW:

This is an excellent review focusing on molecular markers for early detection of advanced neoplasia in IPMNs.

One minor suggestion:

Paragraph: Page 4, line 126 on wards:

The authors need to update the current evidence on risk stratification of IPMNs.

Currently two EUS-guided methods lead the field in risk stratification of IPMNs. These procedures include cyst fluid molecular analysis (alluded to by the authors) and EUS-guided confocal laser endomicroscopy.1-3

Although the followed data are from single-center studies, other multi-center studies are underway.

  • In a large single center study, Singhi et al, showed high accuracy in the detection of IPMNs with advanced neoplasia using molecular markers.4 (This has been cited by the authors under the detailed discussions of molecular markers)
  • In another single center study, EUS-guided confocal laser endomicroscopy image patterns accurately detected IPMNs with advanced neoplasia.5

References:

  1. Krishna SG, Hart PA, Malli A, et al. Endoscopic Ultrasound-Guided Confocal Laser Endomicroscopy Increases Accuracy of Differentiation of Pancreatic Cystic Lesions. Clin Gastroenterol Hepatol 2020;18:432-440 e6.
  2. Singhi AD, Koay EJ, Chari ST, et al. Early Detection of Pancreatic Cancer: Opportunities and Challenges. Gastroenterology 2019;156:2024-2040.
  3. Singhi AD, Zeh HJ, Brand RE, et al. American Gastroenterological Association guidelines are inaccurate in detecting pancreatic cysts with advanced neoplasia: a clinicopathologic study of 225 patients with supporting molecular data. Gastrointest Endosc 2016;83:1107-1117 e2.
  4. Singhi AD, McGrath K, Brand RE, et al. Preoperative next-generation sequencing of pancreatic cyst fluid is highly accurate in cyst classification and detection of advanced neoplasia. Gut 2018;67:2131-2141.
  5. Krishna SG, Hart PA, DeWitt JM, et al. EUS-guided confocal laser endomicroscopy: prediction of dysplasia in intraductal papillary mucinous neoplasms (with video). Gastrointest Endosc 2020;91:551-563 e5.

Author Response

This is an excellent review focusing on molecular markers for early detection of advanced neoplasia in IPMNs.

One minor suggestion:

Paragraph: Page 4, line 126 on wards:

The authors need to update the current evidence on risk stratification of IPMNs.

Currently two EUS-guided methods lead the field in risk stratification of IPMNs. These procedures include cyst fluid molecular analysis (alluded to by the authors) and EUS-guided confocal laser endomicroscopy.1-3

 Although the followed data are from single-center studies, other multi-center studies are underway.

  • In a large single center study, Singhi et al, showed high accuracy in the detection of IPMNs with advanced neoplasia using molecular markers.4(This has been cited by the authors under the detailed discussions of molecular markers)
  • In another single center study, EUS-guided confocal laser endomicroscopy image patterns accurately detected IPMNs with advanced neoplasia.5

R: We thank the reviewer for the valuable comments and appreciated all the suggestions. The references have been included and the section has been updated (line 158).

Round 2

Reviewer 2 Report

The authors addressed the raised issues and improved the article.